# Microfoundations of Data-Driven Antimicrobial Stewardship Policy (ASP)

**DOI:** 10.3390/antibiotics13010024

**Published:** 2023-12-26

**Authors:** Yogita Thakral, Sundeep Sahay, Arunima Mukherjee

**Affiliations:** 1Department of Informatics, University of Oslo, 0373 Oslo, Norway; sundeeps@ifi.uio.no; 2HISP India, New Delhi 110025, India; 3SUSTAINIT—Unit for sustainable health, Faculty of Medicine, University of Oslo, 0316 Oslo, Norway

**Keywords:** ASP, antimicrobial stewardship, AMR, data driven, resource constrained, actors, action, sociomateriality, microfoundations

## Abstract

This paper introduces a comprehensive framework that elucidates the microfoundations of data-driven antimicrobial stewardship programs (ASPs), specifically focusing on resource-constrained settings. Such settings necessitate the utilization of available resources and engagement among multiple stakeholders. The microfoundations are conceptualized as interlinked components: input, process, output, and outcome. Collectively, these components provide a comprehensive framework for understanding the development and implementation of data-driven ASPs in resource-constrained settings. It underscores the importance of considering both the social and material dimensions when evaluating microbiological, clinical, and social impacts. By harmonizing technology, practices, values, and behaviors, this framework offers valuable insights for the development, implementation, and assessment of ASPs tailored to resource-constrained environments.

## 1. Introduction

Antimicrobial resistance (AMR), the ability of microbes to resist the effects of antimicrobial drugs, jeopardizes the ability to treat common infections effectively. However, the burden of AMR is not evenly distributed [1]. Inequities in access to healthcare, sanitation, and antimicrobial agents exacerbate the challenge of AMR. Vulnerable populations, often in low- and middle-income countries (LMICs), face greater challenges in accessing appropriate antibiotics and healthcare services, which can lead to suboptimal treatment, misuse, and overuse of these drugs. Additionally, structural factors like impoverished living conditions, lack of hygiene, and inadequate healthcare infrastructure contribute to the rapid spread of resistant pathogens in these communities [2,3].

Antimicrobial stewardship programs (ASPs) can play a pivotal role in addressing the challenge of AMR by promoting judicious antibiotic use, ensuring that antibiotics are prescribed only when necessary and that patients receive the appropriate treatment for their condition. However, LMICs are burdened with high rates of infectious diseases, which require urgent and competing health priorities that may affect the demand and supply of antibiotics [4]. This leads to a suboptimal use (incorrect dosages, incomplete treatment courses, unnecessary antibiotic use for conditions that do not require them, easy over-the-counter access, poor-quality antimicrobials) of antibiotics, which increases the risk of developing AMR, thereby affecting the health and well-being of those already facing challenges in accessing adequate healthcare resources. This emphasizes the urgent need for measures like ASPs to promote responsible antibiotic use and help mitigate the risks associated with AMR in these vulnerable populations.

However, there is an existing gap related to the entire spectrum of ASP activities in LMICs, from their development as policies to their real-world implications, and this gap demands more attention and investigation to guide the design and implementation of appropriate strategies to combat the challenges posed by AMR in these contexts. LMICs, especially India, are diverse and have historically complex and fragmented health systems that require context-specific adaptation and innovation. Historically, ASPs have been implemented within high-resource settings and not as much in low-resource settings.

Furthermore, existing policies predominantly reflect a biomedical framing at global/national/regional levels, which do not incorporate concerns arising from the social, cultural, and behavioral contexts where people live, environmental exposures, peoples’ experience, access to healthcare, and others, leading to different patient outcomes. Existing research on the effectiveness of antimicrobial stewardship in high-income settings has highlighted racial/ethnic disparities in antibiotic prescribing with lower diagnosis and antibiotic prescribing rates in a limited set of patients [5,6]. However, there is inadequate evidence on the implications of ASPs on mitigating inequities associated with AMR in LMICs. This raises the need to understand the foundations to design and implement data-driven ASPs in an institutionally sustainable manner that is locally relevant with the potential to be scaled to other contexts with both similar and different structural challenges. The role of ASP becomes even more critical in the aftermath of pandemics like COVID-19, which witnessed extensive and often unwarranted use of antibiotics, particularly in LMICs. The urgency to treat potential bacterial co-infections in COVID-19 patients led to an increased demand for antibiotics, contributing to concerns about antibiotic overuse and the subsequent rise of AMR [7]. In LMICs, where healthcare resources may be limited, ASPs become extremely essential to optimize antibiotic prescribing practices, prevent the emergence of drug-resistant strains, and improve patient outcomes.

This paper draws from sociomateriality [8] as a guiding theoretical concept and existing research on developing a health facility-based evidence base in a resource-constrained setting in India to serve as the foundation for evidence-based ASP development. This paper contributes by developing a framework using a data-driven approach for the design and development of a locally relevant ASP. This paper will discuss the foundations of an ASP in terms of the necessary actors, actions, material, and structure involved in the development and implementation of a locally relevant ASP in the next section. In the third section, the framework comprising the components will be discussed, followed by methods and conclusions.

## 2. Antimicrobial Stewardship from a Bottom-Up Approach

Every antimicrobial stewardship program (ASP) is distinct and must be customized to operate effectively within its particular healthcare setting. Nevertheless, ASPs share a fundamental structure comprised of underlying components that should be understood in the context of their development and implementation. To understand and effectively implement ASPs, it is essential to understand these microfoundations while considering the unique context of each program’s development and implementation. This paper explores these dimensions, encompassing the actors participating, the actions they engage in, and an assessment of the impact of the action involved. An interaction between these dimensions will determine the expected and achieved impact of antimicrobial stewardship.

These dimensions are interconnected and interdependent. The outcome and impact of antimicrobial stewardship hinges on how these elements interact. The actors, such as healthcare professionals, pharmacists, physicians, and administrators, work together to execute specific actions, such as prescribing, monitoring, and educating. The material aspects, which encompass drug availability, diagnostic tools, and guidelines, impact the quality and efficiency of these actions. Finally, these components are situated within broader organizational structures, such as existing rules and norms, which are constantly redefined based on the evidence to develop and implement locally relevant ASPs, which further influence their impact.

### 2.1. Actors in Developing and Implementing a Data-Driven ASP

Multistakeholder collaborations are a critical part of developing local and sustainable data-driven ASP. By bringing together various actors associated with different roles and interests, the common goal of improving stewardship practices brings diverse expertise and local perspectives from the end users of the policy toward a more holistic approach to AMS. The end users of the policy include the population that will be ultimately affected by the policy, i.e., at the local level, patients, physicians, community, and healthcare workers. First, we will describe the actors associated with locally relevant ASP development, followed by the role of multistakeholder collaborations.

#### Actors Involved at the Local Level

The local data-driven approach to ASP development must engage in the development and implementation of the perspectives of the users who will be affected by it the most. These actors include hospital management, local antimicrobial stewardship team, and engagement with patients and local communities. A local antimicrobial stewardship team comprising local actors and physicists from all clinical and non-clinical departments in the hospital. Another challenge in locally relevant policy design and implementation is the limited involvement of the wider healthcare workforce, patients, and communities that are potential beneficiaries of an ASP policy [9]. This results in adoption and implementation challenges of the policy, particularly when they come as mandates and guidelines from the top. Often, the process of policy development and implementation does not include the voices, perspectives, and experiences of those affected by them—both patients and communities as recipients of care and healthcare providers as decision makers and in the provision of healthcare.

Historically, the responsibility for treating infections has primarily rested with surgeons, and they need to hold a central position in ASPs and the overall management of infectious diseases. However, the lack of active involvement in policy design undermines the efficacy of their implementation. Additionally, patients and public involvement can potentially provide inputs on lifestyle and preventive aspects of AMR care, which is deemphasized in biomedical approaches. The patients’ worldview about their experiences around access and utilization of healthcare services can widen policy scope to include preventive measures [10]. A bottom-up policy development needs to explore if and how citizen voices can be included in the process of policy development and implementation. Table 1 briefly describes the actors involved and their role in local ASP development. This research will engage with the existing formal and informal groups in the community and understand the perspectives of the patients that can help in strengthening the preventive component of ASPs.

### 2.2. Action in Developing and Implementing a Data-Driven ASP

The action is described in terms of the two broad areas of work required in the development and implementation of data-driven ASPs from a bottom-up approach to ensure they are sustainable: (i) evidence-based data generation work and (ii) multistakeholder engagement and coordinating work. These two broad categories of work form the backbone of the development and implementation of a data-driven antimicrobial stewardship policy. Together, they ensure that the policy is evidence based, tailored to local needs, effectively communicated, and continuously improved to address the challenges of antimicrobial resistance.

These will now be described in detail in the next sections.

#### 2.2.1. Evidence-Based Data Generation Work

Evidence-based data are central to both developing and implementing the stewardship guidelines at all levels. However, a patient-centric evidence base becomes exceedingly relevant at a health facility level that provides actionable guidance for physicians to treat patients. The first step to developing an evidence base requires a baseline assessment as a foundational step in understanding the current status and context-specific needs for policy development, implementation, and monitoring. This will help develop an understanding of the existing conditions, challenges, and resources, enabling factors and stakeholders to help tailor context-specific policies. In addition to identifying the gaps, issues, and opportunities, it will also help to establish benchmarks for evaluating the impact of policy interventions over time. The baseline assessment can help in building a foundation for the process of policy design, which is responsive to local realities and designed to meet the unique requirements of the target population. The assessment provides a foundation for the policymakers with the knowledge and context necessary to design data-driven, context-specific policies.

While the core principles of successful antimicrobial stewardship programs have been widely published [11,12], the process by which they are designed and the data types required for designing a locally relevant policy have received limited attention. ASPs guide healthcare providers in the appropriate use of antibiotics, aiming to optimize patient outcomes while minimizing the development of antibiotic-resistant infections. These policies encompass a range of strategies, including guidelines for antibiotic prescription, infection prevention measures, surveillance of resistance patterns, and ongoing capacity building among healthcare providers [13]. Key challenges in the design and implementation of these policies include a limited understanding of the social and cultural influences in policy design and implementation [2]. An understanding of the social and cultural influences on antibiotic prescription, use, local practices of infection prevention and control, and awareness about the challenge of AMR among different stakeholders can provide relevant information to build a contextually relevant policy.

ASP interventions are typically designed and implemented with limited consideration of the sociocultural and contextual factors that are crucial for their long-term success and sustainability [3]. In most resource-constrained settings, the ASP interventions are led by microbiologists with limited involvement from the wider healthcare workforce [14] and nearly none from citizens. This will include an active component of action where the facilitation of the process of policy development will be performed. The facilitation involves bringing together the stakeholders involved in the process of ASP development to enable communication and dialogue, enabling the provision of relevant data to guide decision making for the process of policy development. The facilitating role will thus be a bridge between the existing knowledge, its integration, and its use in the process of policy development and implementation.

#### 2.2.2. Multistakeholder Engagement and Coordinating Work

Engaging multiple stakeholders, especially the developers, implementers, and those affected by the policy, is pertinent to the development of a locally relevant policy with a bottom-up approach. A bottom-up approach to building an ASP through multiple stakeholder collaboration involves engaging a broad range of actors at the local level and empowering them to contribute to the policy development process. This approach emphasizes grassroots involvement, local expertise, and citizen/community engagement.

In LMICs with weaker public health systems, such collaborations at the health facility level can play a crucial role in developing and implementing antimicrobial stewardship programs. By bringing together multiple stakeholders with diverse expertise, a local health facility-specific policy can be developed by taking into account the needs and requirements of different perspectives and departments in the health facility. This includes an active role and participation from representatives from all clinical and non-clinical departments, in addition to the participation from the facility-specific antimicrobial stewardship team. This requires the assignment of nodal officers or representatives from all departments for both inter and intra-coordination during the policy development and its implementation. These nodal representatives are the people interested in engaging in stewardship activities in their respective departments and playing a leading coordinator role. They play an active and important role and are responsible for coordination with the antimicrobial stewardship committee at the hospital as well as their own departments. These nodal officers identify the key challenges and areas of concern in their departments and, upon discussion with the stewardship team, develop their department-specific guidelines. For example, in public health facilities, especially in LMICs that do not have infectious diseases specialists and clinical pharmacologists, the role of nodal officers from pharmacology and microbiology becomes especially important in coordinating and playing an active role in both the development and implementation of stewardship policies from a bottom-up approach. These nodal officers from different departments develop a working group that meets once a month and organizes individual meetings with nodal officers from each department to take into account individual perspectives from different areas of the health facility and organize collaborative activities such as prescription audits, feedback, and monitoring antimicrobial use throughout the hospital.

Crafting an AMS policy involves designing and structuring the policy framework based on the data and research findings. This work includes defining policy objectives, strategies, guidelines, and protocols for antimicrobial use. It requires the input of experts, healthcare professionals, and policy designers to ensure the policy is evidence based and aligns with the local context. Once the policy is developed, there is a need for extensive work related to its implementation. This includes coordinating efforts among various stakeholders, healthcare institutions, and local communities to ensure that the policy’s guidelines and recommendations are put into practice. It may involve setting up coordination committees, training programs, and communication strategies to facilitate policy implementation.

### 2.3. ASP Impact Assessments

ASPs are quality-motivated programs aimed at improving the use of antibiotics in healthcare facilities. Existing research indicates that between 25 and 50% of hospitalized patients receive antibiotics, with between 30% and 50% of antibiotic use being inappropriate [15]. The primary goal is to optimize clinical outcomes, minimize adverse consequences, and increase resistance [16]. However, challenges exist in identifying the impact of these interventions because of a lack of the appropriate design to adequately assess the success or failure of ASP interventions or evaluation, which limits an understanding of the impact of stewardship policies [17].

To gauge the impact and effectiveness of the data-driven AMS policy, continuous monitoring and evaluation work is crucial. This involves ongoing assessment of antimicrobial use practices, resistance trends, and healthcare outcomes. Feedback loops are established to identify areas that require improvement and to make necessary adjustments to the policy. It also includes compliance monitoring and the use of metrics and key performance indicators to track progress. These can be further categorized into (i) microbiological and clinical outcomes and (ii) social outcomes that the ASPs developed using a bottom-up approach must aim to achieve. These are further described below.

#### 2.3.1. Microbiological and Clinical Outcomes

Microbiological and clinical outcomes are related to factors that have a direct impact on patients’ health, such as the reduction in antibiotic-resistant infections, improved patient recovery, and the effective treatment of infectious diseases. These can be further classified into short-term and long-term effects.

Short-term outcomes of ASP implementation include immediate changes and impacts observed after the implementation of stewardship interventions. These outcomes include shifts in the prescribing practices of physicians, often leading to improved adherence to established guidelines for antibiotic use. Short-term results also include changes in the clinical outcomes of patients, such as a reduction in hospital readmissions and a decrease in the incidence of hospital-acquired infections. These early outcomes are important in reflecting the policy’s initial effectiveness and its ability to bring about immediate improvements in healthcare practices and patient care.

Long-term outcomes of ASPs include the sustained effects and lasting benefits that emerge over an extended period. These outcomes focus on the more long-term implications of the policy. They include the assessment of treatment outcomes, such as mortality rates, which can showcase the policy’s ability to save lives and reduce the severity of infections. Additionally, long-term outcomes involve monitoring changes in resistance patterns among priority pathogens. A successful ASP must also lead to a decline in antibiotic-resistant strains, ultimately protecting patients from difficult-to-treat infections. Furthermore, long-term outcomes encompass the assessment of the length of hospital stays, which, when reduced, indicates a more efficient and cost-effective healthcare system. Collectively, both short-term and long-term outcomes serve as valuable measures to gauge the overall impact of data-driven AMS policies, ensuring that they fulfill their intended objectives of improving patient care and mitigating antimicrobial resistance.

#### 2.3.2. Social Outcomes

Beyond clinical outcomes, ASPs developed through a bottom-up approach should aim to achieve social outcomes. This includes fostering responsible antimicrobial use, raising awareness about the importance of stewardship, and encouraging community engagement [18]. These social outcomes are essential for the long-term sustainability of AMS policies, especially in resource-constrained settings where the social dimension of AMS policies takes on a critical role in ensuring the long-term sustainability and success of such initiatives.

Short-term social outcomes involve immediate changes in citizens (patients and community engagement), behaviors, and awareness regarding antibiotic use. In the short term, these programs aim to educate and engage communities on responsible antibiotic usage, raise awareness about the dangers of antibiotic resistance, and foster a sense of collective responsibility. Changes in public perception and behavior can be assessed through observations, interviews, discussions, and community participation in AMS activities.

Long-term social outcomes, however, delve into the sustained impacts of these ASPs. They include lasting changes in antibiotic usage behaviors, a reduction in unnecessary antibiotic demand and prescription, and a shift toward responsible self-care and healthcare-seeking practices. Long-term outcomes also involve a decrease in self-medication with antibiotics, which is a common practice in many LMICs. These outcomes promote a culture of prudent antibiotic use and empower communities to play an active role in their health. Long-term assessment of these outcomes may require extended follow-up studies and community-based interventions. The measurement of these social outcomes necessitates a robust and context-specific approach. It involves the collection of data on changes in knowledge, attitudes, and practices related to antibiotic use among the public and communities. Interviews, focus groups, and educational campaigns can be used to gauge the impact of ASPs on community engagement and antibiotic use behaviors. Additionally, ongoing monitoring and evaluation are essential to track the progress of these social outcomes over time, as it often takes years to observe significant changes in behaviors and awareness.

Overall, in LMICs, the success of data-driven ASPs developed from a bottom-up perspective relies on their ability to not only improve clinical outcomes but also drive essential social changes. By facilitating a deeper understanding of antibiotics, promoting responsible use, and engaging communities, these programs must aim to create sustainable practices that combat antimicrobial resistance and improve public health in the long run.

## 3. Sociomateriality in ASP Development and Implementation

Sociomateriality refers to the constitutive entanglement of the social and the material in everyday organizational life [19]. Material is a term that refers to the physical and tangible aspects of the world, such as objects, bodies, spaces, and technologies. Material is contrasted with social, which refers to the human and cultural aspects of the world, such as beliefs, norms, values, and practices. However, material and social are not separate or independent domains; rather, they are entangled and mutually constitutive. This means that material things shape and are shaped by social meanings and actions, and vice versa.

In the context of antimicrobial stewardship, the material encompasses antibiotics, diagnostic tools, surveillance systems, monitoring and evaluation systems, and other physical resources crucial for the management of antimicrobial use. These material components are essential in the practical execution of the ASP and directly influence how healthcare professionals make decisions regarding antibiotic prescribing and patient care. The social aspects encompass the attitudes and behaviors of healthcare professionals regarding antimicrobial use, the cultural norms within the healthcare facility, and the shared values related to responsible antibiotic prescription and patient education. This means that the way material resources are used and managed (e.g., antibiotics) is heavily shaped by the social interactions, beliefs, and practices of healthcare professionals. Similarly, the social context of an organization, including the formal rules, cultural norms, and values, are significantly influenced by the material resources and technologies available for antimicrobial stewardship.

Effective policy development and implementation require understanding how the material (e.g., antibiotic availability, availability of an evidence base, data systems to enable decision making) and the social (e.g., healthcare professionals’ prescribing practices and citizen engagement) interact and influence each other. Acknowledging the entanglement of these dimensions is crucial for creating data-driven ASPs that are not only technically sound but also culturally and socially relevant. For instance, the implementation of a data-driven ASP may involve the introduction of a digital monitoring system for data analysis (a material element). However, the effectiveness of this software relies on healthcare professionals’ willingness and ability to utilize it, understand the insights it provides, and incorporate those insights into their practices (social elements). Conversely, healthcare professionals may request specific data to inform their decisions, which necessitates the availability of data systems and technology (material elements) to provide the required information. Understanding sociomateriality in the context of ASP development and implementation highlights the need to consider both the social and material dimensions in a holistic manner. It underscores that the sustainability of antimicrobial stewardship programs depends on aligning technology, practices, values, and behaviors, all of which are interconnected in a dynamic and evolving healthcare ecosystem. This approach can lead to more effective data-driven ASPs that are attuned to the complexities of the healthcare environment and its multidimensional interactions. Guided by this approach, the framework in Figure 1 describes the microfoundations of ASPs in resource-constrained settings.

The microfoundations are conceptualized as interlinked components: input, process, output, and outcome. These components collectively provide a comprehensive framework for understanding the development and implementation of data-driven ASPs in resource-constrained settings. Inputs represent the initial building blocks required for the development of a data-driven ASP.

These inputs describe the diverse range of actors involved, including healthcare professionals, administrators, and local antimicrobial stewardship teams. The process discusses the network of stakeholders and their multifaceted engagement within the ASP development process. It highlights the adaptive nature of this policy development, emphasizing the collaborative efforts of multiple stakeholders. This process is dynamic and iterative, leading to tangible outputs. Outputs serve as the concrete manifestations of the ASP development process. These outputs include the establishment of feedback loops, the formulation of guidelines, and other tangible elements that result from the collaborative efforts of the stakeholders. The outcomes of the ASP development process encompass the microbiological and clinical impacts on healthcare practices, as well as the broader social consequences. These outcomes are multifaceted and can be further assessed in terms of short-term and long-term effects. By acknowledging the sociomaterial nature of ASPs, this framework becomes a robust tool for understanding the complexity of antimicrobial stewardship in resource-constrained settings and underscores the importance of addressing both social and material dimensions for effective policy development and impact assessment.

### Role of Digital Technology in ASP Development and Implementation

Digital technologies, including surveillance, monitoring and evaluation systems, electronic digital records, and hospital information systems, can play a crucial role in the development of ASPs by providing tools for evidence-based decision making. These technologies represent tangible components within the broader material aspects of ASP, influencing the local patient care processes. However, in resource-constrained settings, public health facilities struggle with proprietary health information systems with limited capacity and resources to maintain them. There are challenges with sub-adequate data and the gap between data production and data use in health information systems in resource-constrained contexts [20]. Data production refers to the processes and activities involved in collecting, recording, transmitting, and storing data from various sources and levels in the health system. Data use refers to the processes and activities involved in analyzing, interpreting, disseminating, and applying data for various purposes and audiences in the health system. The gap between data production and data use is caused by several technical and sociocultural factors that affect data quality and use [20].

In the case of AMR and ASP, these factors include the availability, accessibility, reliability, and compatibility of the hardware, software, and network infrastructure that support data collection, storage, and analysis. Technical factors also include the design, functionality, and usability of the applications. One of the main barriers to adopting digital technologies for AMR in resource-constrained settings is the cost of its purchase and maintenance, which highlights the open-source approach as a suitable solution for resource-constrained areas [21]. The monitoring platforms to capture and analyze data for AMR, which is a starting point for ASP development developed using proprietary sources, have presented challenges like lack of system interoperability and lack of data standards.

Local practitioners encounter several challenges in the implementation of global platforms like WHONET for AMR surveillance. Additionally, the Global Antimicrobial Resistance Surveillance System (GLASS) falls short in providing patient-based information at the hospital level and lacks essential clinical metadata related to antimicrobial use and the duration of hospitalization [22]. These data are the backbone of developing an evidence-based ASP. These limitations pose obstacles to gaining a comprehensive understanding of AMR dynamics within the local healthcare context. Furthermore, there are no existing mechanisms to generate routine data on antibiotic use and their comparisons with the resistance patterns. The challenges extend beyond technical issues and also include cultural barriers, a lack of experience, and the need for context-specific solutions to align with global standards while addressing the unique needs at the practice level. Overcoming these challenges requires a nuanced approach that considers both technical and cultural aspects, ensuring the strategies to design, develop, and implement digital technologies are tailored to local contexts while meeting global standards [23].

## 4. Materials and Methods

The methodology for identifying the microfoundations of ASP is primarily based on a substantial experience garnered from building and implementing an AMR monitoring system over the past four years. This methodology is also informed by ongoing work in the field of AMR, spanning primary, secondary, and tertiary healthcare organizations, as in building ASPs at the public healthcare facility level in India. This combines a theoretical foundation, practical experience, and ongoing work on AMR in resource-constrained public settings in India.

### 4.1. Sociomateriality in ASP Development and Implementation

Sociomateriality [8] refers to the entwined and mutually constitutive relationship between the social and material dimensions within everyday organizational life. In the context of ASPs, the “material” aspect encompasses the tangible components, such as medications, diagnostic tools, and guidelines, while the social aspect pertains to the human and cultural elements, including beliefs, norms, values, and practices. However, the essence of sociomateriality lies in recognizing that these two dimensions are not isolated domains; instead, they interact and shape each other continuously. This study draws from sociomaterialty to identify the microfoundations of ASPs by exploring the interactions between actors, their actions, and the material components within healthcare settings.

### 4.2. Practical Experience

The methodology draws significantly from the firsthand experience of building and implementing an AMR monitoring system over the course of four years. This hands-on experience provided valuable insights into the challenges, strategies, and outcomes associated with real-world ASPs and data-driven policy development [24]. It also informed the text’s approach to addressing the practical aspects of implementing AMS policies.

### 4.3. Ongoing Work in Public Healthcare Organizations

In addition to historical experience, ongoing work in primary, secondary, and tertiary healthcare organizations was considered. This work involves the continuous assessment of antimicrobial use practices, resistance trends, and healthcare outcomes. Feedback loops have been established within these healthcare settings to identify areas that require improvement and to make necessary adjustments to the policies [25]. A critical element of the methodology is the active engagement of multiple stakeholders. The involvement of healthcare professionals, administrators, and pharmacists reflects the bottom-up approach. A coordinated effort has been undertaken to facilitate communication, knowledge exchange, and the integration of various perspectives in policy development and implementation.

## 5. Conclusions

Drawing from sociomateriality, this paper has developed a framework highlighting the microfoundations of data-driven ASPs that are especially relevant for resource-constrained settings. Resource-constrained settings, especially in LMICs with limited resources and limited manpower engaged in stewardship activities, require the usage of existing resources and engagement of multiple stakeholders in first developing an evidence base and then further using the evidence base for the development of policy and assessing its impact based on the baseline identified. The framework highlights that antimicrobial stewardship is complex, especially in resource-constrained contexts, and always changing. It stresses the importance of looking at both the social and material aspects of microbiological, clinical, and social impact by aligning technology, practices, values, and behaviors.

## Figures and Tables

**Figure 1 antibiotics-13-00024-f001:**
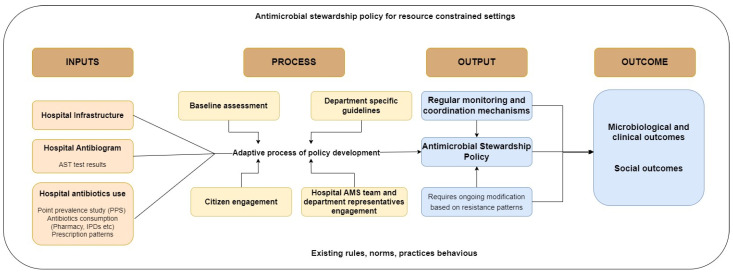
Microfoundations of a data-driven ASP for resource-constrained settings.

**Table 1 antibiotics-13-00024-t001:** Actors and their role in ASP development.

Actors	Role in Local ASP Development
Hospital Management	- Provide leadership and support for a locally relevant ASP approach
	- Allocate resources for the development and implementation of the ASP policies
Local Antimicrobial Stewardship Team	- Comprising local actors and physicists from clinical and non-clinical departments
	- Actively participate in the development and implementation of ASP policies
	- Collaborate with hospital management to ensure alignment with local needs
Patients and Local Communities	- Insights into firsthand experiences and expectations
	- Represent the community’s perspective to widen the policy scope to include preventive measures
Surgeons	- Hold a central position in ASPs and infectious disease management
	- Actively participate in policy design to ensure their perspectives are considered
Healthcare Workforce	- Ensure active involvement in policy design and implementation processes
	- Provide insights into the challenges and opportunities in adopting ASP policies
Patient Advocacy Groups	- Represent patient perspectives on access and utilization of healthcare services
	- Advocate for policies that address the needs and concerns of patients
Research and Academic Partners	- Conduct research to inform the development of evidence-based ASP policies
	- Collaborate with local stakeholders to integrate research findings into policies

## Data Availability

The data presented in this study are available on request from the corresponding author.

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
