# Peer review of "Microfoundations of Data-Driven Antimicrobial Stewardship Policy (ASP)"

_antibiotics, 2023, doi:10.3390/antibiotics13010024_

Round 1

Reviewer 1 Report

Comments and Suggestions for Authors

It is a detailed, well written and well researched paper on strategies to curb Antimicrobial resistance in LMIC countries.

It is very wordy. Could there be tables where individuals and their roles are mentioned. That would be easier for the reader to understand.

What major challenges to the authors foresee. 

The authors can also suggest some changes to existing Health Information systems/ Electronic Medical Records and their role in curbing AMR. 

Author Response

In this version of the manuscript, as suggested by the reviewer:

 - We have added a table to describe the actors involved and their roles in the process of policymaking

 - Added a new section to describe the role of digital technology in AMR and antimicrobial stewardship in the discussion section

 - Have elaborated the methods section

 - Have made minor language corrections

Reviewer 2 Report

Comments and Suggestions for Authors

Dear Authors,

I have one suggestion:

- to discuss how health crisis (e.g. COvid-19 pandemic or wor etc.) could influence the ASP. Maybe, we have to develope ASP in this specific situation, too. 

Author Response

As suggested by the reviewer, we have tried to bring in the role of antimicrobial stewardship during pandemics and especially in the aftermath of Covid-19 briefly in the introduction section. 

Reviewer 3 Report

Comments and Suggestions for Authors

Title

The authors' goal appears to be to identify microfoundations that are useful for establishing data-driven Antimicrobial Stewardship Programs (ASPs), particularly targeting healthcare facilities in resource-limited settings.

Introduction

During the development of this section, the authors describe the concepts necessary to understand the topic and why the strategies implemented in environments with limited resources have not been successful.

Materials and Methods

The authors' starting point for this section is based on a practical experience of four years of development at the different levels of care of the Indian health services.

Results

The authors present in a well-structured and illustrative manner the microfoundations for establishing data-driven Antimicrobial Stewardship Programs in resource-limited settings.

Conclusion

It establishes the importance of having an evidence base to be able to develop a strategy that will generate a positive impact in both the social and material aspects.

Author Response

Thank you for the positive comments on the paper.